# The Association Between Affective Temperament Traits and Dopamine Genes in Obese Population

**DOI:** 10.3390/ijms20081847

**Published:** 2019-04-15

**Authors:** Natalia Lesiewska, Alina Borkowska, Roman Junik, Anna Kamińska, Joanna Pulkowska-Ulfig, Andrzej Tretyn, Maciej Bieliński

**Affiliations:** 1Chair and Department of Clinical Neuropsychology, Nicolaus Copernicus University in Toruń, Collegium Medicum, Bydgoszcz 85-094, Poland; alab@cm.umk.pl (A.B.); joanna.pulkowska@gmail.com (J.P.-U.); bielinskim@gmail.com (M.B.); 2Department of Endocrinology and Diabetology, Nicolaus Copernicus University in Toruń, Collegium Medicum, Bydgoszcz 85-094, Poland; junik@cm.umk.pl (R.J.); amikam@wp.pl (A.K.); 3Department of Biotechnology, Nicolaus Copernicus University, Toruń 87-100, Poland; prat@umk.pl

**Keywords:** dopaminergic gene polymorphisms, affective temperament, obesity

## Abstract

Studies indicate the heritable nature of affective temperament, which shows personality traits predisposing to the development of mental disorders. Dopaminergic gene polymorphisms such as *DRD4*, *COMT*Val158Met, and *DAT1* have been linked to affective disorders in obesity. Due to possible correlation between the aforementioned polymorphisms and the affective temperament, the aim of our research was to investigate this connection in an obese population. The study enrolled 245 obese patients (178 females; 67 males). The affective temperament was assessed using the Temperament Evaluation of Memphis, Pisa, Paris, and San Diego autoquestionnaire (TEMPS-A). Genetic polymorphisms of *DAT1*, *COMT*Val158Met and *DRD4* were collected from peripheral blood sample and determined using a polymerase chain reaction (PCR). Only in COMT polymorphisms, the cyclothymic and irritable dimensions were significantly associated with Met/Val carriers (*p* = 0.04; *p* = 0.01). Another interesting finding was the correlation between the affective temperament and age in men and women. We assume that dopamine transmission in heterozygotes of COMT may determine the role of the affective temperament in obese persons. Dopaminergic transmission modulated by COMT may be responsible for a greater temperament expression in obese individuals. To our knowledge, this is the first study describing the role of affective temperament in the obese population, but more research is needed in this regard.

## 1. Introduction

Previous research devoted to eating disorders, mainly related to anorexia and bulimia, indicated the possibility of specific personality traits related to both the predisposition to the disease and those affecting the course and clinical picture of the disease [1]. The psychological aspects of predisposition to obesity are mostly: disorders of the self-regulation mechanism, beliefs and expectations of the individual, personality traits, difficulties in coping with stress and experienced emotions [2]. Recent psychiatric studies suggest that there is a link between obesity and mood disorders. The association between obesity and depression occurred in childhood. Previous research indicated that the symptoms of eating disorders are common and that patients with bipolar disorder are more obese than the control group [3,4,5]. The results indicate that the symptoms of eating disorders are common and that patients with bipolar disorder are more obese than the control group [6,7]. Along with the broadening of the limits of diagnostic criteria for bipolar disorder (BD) over the last years, research has pointed to the high prevalence of less severe forms of BDs, in particular hypomania, among obese patients [8]. Dopamine might be a factor linking obesity with mood disorders, especially given that maladaptive changes in dopaminergic transmission have been observed in obesity and [9,10,11].

Yokum et al. (2015) tested the multilocus genetic composite risk score—a proxy for dopaminergic signaling—and future changes in BMI values. The results of their study revealed that *DRD4*, *COMT*Val158Met and *DAT1* polymorphisms, putatively associated with a greater DA signaling capacity, were linked to greater increases in the BMI; hence the future weight gain [12].

According to the regulatory theory, the temperament is the basic, relatively permanent character traits that manifests in the formal specifics of behavior. These features are already present in early childhood and are common to humans and animals. Being originally determined by innate physiological mechanisms, temperament may change under the influence of puberty, aging and certain environmental factors. In their work, Serafini et al. (2015) showed that unpleasant events, inter alia: sexual abuse, physical abuse, child maltreatment or domestic violence, were associated with greater depression and suicidality in adolescents. It is worth noting that the type of events, as well as the frequency and the timing of maltreatment, may influence the risk of psychiatric disorders, including suicidal behavior, due to the disruption in the brain development connected to cognitive, social or emotional functioning [13].

According to Arnold Buss and Robert Plomin (1984), temperament is a set of inherited personality traits that are revealed in early childhood. The temperament understood in this way is the basis for shaping and developing personality [14]. According to the assumptions of modern psychiatry, temperament is considered a personality aspect that takes into account the constant behavior of the individual, predicts mood changes and is strongly genetically conditioned [15,16,17]. 

An important researcher in the field of psychiatry, Emil Kraepelin, believed that a depressive temperament, and a manic, irritable and cyclothymic temperament is not only represented by affective predispositions, but also by subclinical variations of manic and depressive disorders. Akiskal et al. distinguished four types of affective temperament: depressive, manic, irritable and cyclothymic. In later studies, manic temperament was changed to hyperthymic temperament, and anxiety temperament was added [18,19,20]. The conceptualization of these five types of temperament has led to the creation of a TEMPS psychometric tool (Temperament Evaluation of Memphis, Pisa, Paris and San Diego). In studies utilizing this tool, obese patients showed significantly higher results in cyclothymic, irritable and anxious temperaments compared to the control group [21]. Assuming that the cyclothymic temperament is part of the mild spectrum of BD, these results are consistent with previous studies suggesting a higher incidence of bipolar symptoms in people with obesity [8].

The relationship between temperamental traits in Cloninger’s concept (Temperament and Character Inventory—TCI) and gene polymorphism for the serotonergic and dopaminergic systems was also found. Research is still under way to determine the role of genes in the regulation and emergence of bipolarity and affective temperament [22]. So far, in obesity, this type of research is scarce. Our previous study showed a significant contribution of the SERT gene in the regulation of temperament in the obese population [23]. 

There are few studies in the literature describing the connection between polymorphisms of the dopaminergic system genes with personality traits, character or temperament. Thus, the aim of this project is to determine the possible role of dopaminergic pathways in the regulation of the affective temperament in the obese population. In order to accomplish our objects, we formulated the following hypotheses:

1. Individuals with higher BMI values will have greater scores in cyclothymic, anxious and irritable temperaments, which are associated with the predisposition to psychiatric comorbidities.

2. A lower dopaminergic transmission modulated by the following gene polymorphisms: *COMT*Val158Met, *DRD4* and *DAT1,* will be associated with a higher BMI and more pronounced cyclothymic, anxious and irritable dimensions.

## 2. Results

Basic demographic data and TEMPS-A dimensions in a group of women and men are shown in Table 1. There were significant differences only in terms of more depressed and irritable dimensions in the group of men.

Table 2 shows the analysis of associations between the temperamental dimensions (according to TEMPS-A) and both the age and BMI. Our results revealed, that in the group of women, a greater age significantly correlated with more expressed dimensions of depression and anxiety. Regarding the BMI, we observed its positive correlation with a greater expression of the hyperthymic temperament and a smaller cyclothymia. On the other hand, in the group of men, there was a negative correlation between age and cyclothymia. In this group, the dimensions of cyclothymia and irritability were significantly more pronounced, as the BMI values increased. A partial Kendall’s regression in the group of women showed the significance of the relationship between the age and depressive temperament, the age and anxiety temperament as well as between the BMI and cyclothymic temperament. On the other hand, in the group of men, the significance was confirmed for the BMI and cyclothymic temperament.

When analyzing the correlations of the studied COMT gene polymorphisms in the subgroups of both sexes (Table 3), no significant relationships were found. Thus, we performed an ANOVA for the entire group, and then conducted a post hoc analysis only for significant results for the ANOVA, which revealed a significantly greater expression of cyclothymia in the heterozygote subgroup. Similarly, the irritability was more pronounced in the heterozygous group.

A multiple testing procedure was then performed to confirm the validity of the relevant results. After applying the Bonferroni correction, it was confirmed that the still results for the COMT gene alleles and TEMPS-A cyclothymic (*p* = 0.01) and irritable (*p* = 0.01) dimensions are considered to be significant.

According to Table 4, the analyses carried out for the *DAT1* polymorphism did not show any significant relationships of temperament dimensions according to TEMPS-A.

Due to a small group of *DRD4* L/L carriers, we combined groups of individuals with L/L and L/S together, tagged them as L-carriers, and performed proper calculations. Nevertheless, as shown in Table 5, the obtained results regarding the analysis of the dependencies for DRD4 polymorphisms did not show any significant associations, in the examined group of obese subjects.

After making calculations of one-dimensional analyses on TEMPS-A (Table 6), we confirmed the significant interaction effect for the gender and following temperaments: depressive, cyclothymic and irritable; for the BMI and anxious temperament; and for *COMT* Val158Met and both the cyclothymic and irritable temperament. However, we did not observe any significance for the age and other examined polymorphisms (Table 6).

The Wald statistic in the logistic regression model indicated the coefficient of gender to be a significant predictor of the TEMPS-D results, and the *COMT* polymorphism to be a significant predictor of the TEMPS-H and TEMPS-I results (Table 7). These test results for *COMT* in predicting TEMPS-C and TEMPS-D, and for *DAT1* in predicting TEMPS-I, remained in the trend.

## 3. Discussion

To date, many studies point to the connection between obesity and mood disorders, such as depression or BD [24,25,26,27,28]. Oniszczenko et al. (2015) suggest that personality traits expressed by temperament may constitute specific risk factors for the development of obesity. Those traits might determine behaviors which hinder weight loss or cause excess eating. Moreover, mentioned temperaments may also contribute to the proneness to mood disorders associated with obesity [29]. 

Therefore, research on a neurobiological basis of affective temperament could convey essential details of how dopaminergic gene polymorphisms add to the pathogenesis of mood disorders in the obese population; it may, in particular, explain that changes in dopamine transmission may be a causative and a common factor in the development of obesity, as well as of affective diseases [30,31,32]. 

In this study, we analyzed affective temperament dimensions in an obese population using the TEMPS-A autoquestionnaire. Subsequently, we scrutinized correlations of affective temperament and dopaminergic gene polymorphisms which are involved in obesity and mood disorders. Those genes are comprised of *COMT* Val158Met, *DAT1* and *DRD4*. To our knowledge, this is the first study analyzing the affective temperament in the context of dopaminergic genes in an obese population.

Table 1 and Table 2 show significant differences of affective temperament dimensions in both sexes. According to Table 1, men scored higher than women for the depressed and irritable temperament. The logistic regression model (Table 7) shows significant results for gender and TEMPS-D, but not for the irritable dimension. In our previous study, evaluating the affective temperament in an obese Polish population in the context of the serotonin transporter gene polymorphism (5-HTTLPR), we also observed a higher expression of the irritable temperament in men [23]. Studies show significant differences between temperament dimensions in patients suffering from BDs in comparison to healthy ones. Individuals with BD show greater scores in depressive, cyclothymic, irritable and anxious dimensions [33]. It has been shown that, among bipolar patients, cyclothymic and irritable temperaments may be connected with impulsivity [34]. The French study of Bénard et al. (2017) exhibits a stronger association between impulsivity and obesity in men than in women, suggesting the role of gender in weight status and eating behaviors [35]. Such results are interesting in the context of the proneness to affective disorders in this population, with a differentiation between both sexes.

The literature also shows that females may be more susceptible to depression than men [36]. This may stem from many factors, including sociocultural, psychosocial, or behavioral factors. Considering the molecular basis which connects gender, depression and obesity, the difference in sex hormones may affect a response to stressors and modulate immune responses, resulting in higher inflammation, eventually leading to depressive disorders [37,38,39]. Sex hormones affect the immune system by exerting pro-, or anti-inflammatory effects. This includes stimulating the immune cell activation, or an increased expression of cytokines which participate in the immune responses. Great evidence points to the link between elevated pro-inflammatory cytokines and depression. The data indicate that the immune system may contribute to depression pathogenesis in different ways due to sex differences. During puberty, a crucial period for depression development, the estradiol levels increase. Also, the interplay between sex hormones and the immune system may be seen in peri- and post-partum depression, where the level of estrogen is also augmented [40]. Androgens take part in the suppression of immune responses, but it has been shown that a greater activation of the immune system in males with a reduced testosterone concentration may contribute to mood disorders [41]. Even though the literature shows mixed results in this field, Byrne et al. (2015) conclude that the female sex may be the factor influencing immune responses and depression [38,42]. More research focusing on differences of affective temperament in both sexes would bring interesting data regarding the genetic and molecular basis of morbidity for mood disorders in men and women. 

Affective temperament is considered a stable construct associated with genetic transmission and could serve as a phenotype to detect genes responsible for a susceptibility to affective disorders [18,43,44]. Surprisingly, we have observed the correlations between temperament dimensions and age in both men and women. A positive correlation between a depressive and anxious temperament and age may ensue from changes of a person’s experience during their lifetime. The study of Caserta et al. (2011) showed no connections between depression and the immune system in young girls, although in older girls higher depression measures were associated with increased NK cells cytotoxicity [42]. It has been shown that a positive demeanor, i.e., extraversion, agreeableness, or being optimistic, may affect the immune system, by for example lowering the IL-6 response to the stress factors [45,46]. On the other hand, pessimism contributed to augmented markers of inflammation, like IL-6 and the C reactive protein [47]. We assume, that similar associations might be responsible for our results regarding TEMPS-A, and that sex and age might constitute potential modifiers of affective temperament dimensions. Furthermore, more research should be conducted in relation to the association between anxiety and depressive disorders, in the context of hypothalamic-pituitary-adrenal (HPA) axis dysregulation [48,49]. 

Epigenetics is a novel field describing alterations in gene functioning without changes within the genome sequence. It provides potential mechanisms explaining the adverse effects of environmental factors on modulatory mechanisms of gene expression, which may exert long-term effects and be putatively heritable [50,51]. Recent studies connect epigenetic changes with numerous diseases including cancer, while laying emphasis on their crucial role in the pathopsychology, by explaining the association between depressive and anxiety disorders, and adverse life events, or the impact of stress in childhood [52,53,54,55,56]. Additionally, in some studies, it has been corroborated that epigenetic changes may exert dysregulations in the HPA axis, by affecting its regulatory genes, thus contributing to stress-related disorders. The upregulation of the cortocotropin-releasing hormone expression or altered transcription of the glucocorticoid receptor in the brain regions may stem from stress-induced epigenetic modifications, and thus be responsible for HPA-axis dysfunction [57,58].

Therefore, we assume that epigenetics might be a putative link connecting received TEMPS-A results and age. Due to the scarce literature regarding this topic, we encourage more research engaged in psychoneuroimmunology or the influence of environmental factors on the affective temperament. Epigenetics constitutes a challenging field which may convey essential data explaining discrepancies in affective temperament investigations.

In the current study, an increased BMI positively correlated with a greater expression of hyperthymic temperament in women and a greater cyclothymic and irritable dimension in men. We can refer to our findings from our previous study. Temperament results between morbid obese (BMI > 40) and obese individuals (BMI ≤ 40) showed that morbidly obese scored greater in hyperthymic and cyclothymic dimensions [23]. In the study of Amann et al. (2009), patients with morbid obesity displayed higher scores in cyclothymic, irritable and anxious dimensions, which is partially consistent with our results [21]. Considering that studies show associations between the cyclothymic, irritable and hyperthymic temperament, and BD, the abovementioned data imply a heightened risk of this disease with a weight gain in obese patients [59,60,61,62]. In this study, the cyclothymic temperament in women showed a negative correlation with the BMI and with the age in males, which is inconsistent with findings in the literature [63]. We presume that the heterogeneity of the results may stem from the lack of the control group. It is possible that, when comparing with non-obese individuals, the study group could exhibit a more expressed cyclothymic dimension of the affective temperament. 

The association between *COMT* Val158Met and mood disorders has been pointed out in the literature [64,65,66]. However, many researchers still show some concerns about the exact mechanism by which dopamine transmission, determined by *COMT,* contributes to the origin of affective disorders [67]. Some authors propose that the polymorphisms may influence the HPA axis reactivity and thus, by causing a dysregulation of the inflammation processes, may be involved in the pathogenesis of mood disorders and obesity in a reciprocal manner [68,69,70]. The literature also shows an association between *COMT* polymorphisms and personality traits in patients suffering from BD [71,72,73].

Some publications exhibit connections between Met alleles and vulnerability to stress and anxiety, and thus depression [65,74]. However, Massat et al. (2011) showed that the Val allele was more common in individuals with an early stage of depression [75]. The study performed on larger population showed mixed results: The Met allele occurred less frequently among men with depression in comparison to the control group [76]. 

During the analysis of the connection between affective temperament and dopaminergic gene polymorphisms, we have only observed the association between *COMT*Val158Met polymorphisms. Considering the affective temperament, *COMT* heterozygotes showed significant results only in irritable and cyclothymic dimensions. Using a logistic regression model (Table 7), we also received significant results concerning the irritable temperament and *COMT* polymorphism. Both temperaments were overrepresented in patients with bipolar disorders [59]. The irritable temperament has been linked with anxiety and agitation and found more often in persons with bipolar disorder, in comparison to healthy controls or patients with a major depressive disorder [62,77]. 

Previous studies on the COMT relationship with the dimensions of the temperament in Cloninger’s concept were focused mainly on the novelty seeking dimension. These studies gave different results, the majority of which focused on the polymorphism rs4680 [78,79,80]. Golimbet et al. (2007) provided evidence that the *COMT* Met allele (which contributes to the reduction of enzyme activity and ultimately leads to an increase in dopamine levels) was associated with a greater severity of temperamental trait novelty seeking in women [78]. The repetition of this result was done by Tsai and co-workers (2004) on young Chinese women [81]. However, the association of the rs4680 polymorphism of the *COMT* gene with the novelty seeking dimension of temperament has not been confirmed. Searching in other studies conducted on the Caucasian population and the Japanese population [79,80]. In a study conducted on the Chinese population on drug addicts, the *COMT* gene polymorphism was shown to be related to the temperamental characteristics of novelty seeking and the tendency to addiction [82]. A decreased pre-dopaminergic activity and low control, associated with specific *COMT* genotypes, may increase impulsivity, which is a component of novelty seeking. Research by Kang and co-workers (2010) on the dimensions of character showed that the Val158Met *COMT* polymorphism may be related to a susceptibility to boredom and the need for strong sensations in women [83].

The TEMPS-A validation study showed a positive correlation between both the cyclothymic and irritable temperament and the higher novelty seeking scores; hence, our findings are consistent with the results of the abovementioned studies [84], in particular in relation to the fact that Cloninger’s novelty seeking, as well as Akiskal’s cylothymic and irritable dimension, are involved in affective disorders [85]. In their work, Parneix et al. (2014) found that patients with irritability related to major depressive episodes were characterized with atypical features like weight gain and showed greater novelty seeking. The authors suggested that such findings are indicative of a greater vulnerability to BD [86]. In another study, impulsivity seen in the bipolar spectrum was also described in the context of obesity and food addiction [87]. Thus, the affective temperament seems to be related to a susceptibility to mood disorders in obese individuals, and its evaluation might provide useful information considering treatment approaches.

Unfortunately, due to the observational design of our study and the lack of a control group, it is difficult to explain the molecular basis of the interplay between the dopaminergic transmission modulated by *COMT* and the affective temperament. We speculate that obese individuals, in comparison to healthy persons, show a disturbed dopamine transmission, and that dopaminergic signaling in heterozygotes gives rise to more pronounced affective temperament dimensions. This may constitute the link between *COMT* polymorphisms and affective disorders in the obese population. Moreover, individual changes in the dopaminergic transmission might bias the obtained results and influence the temperament expression or exert differences in one’s behavior [88,89]. We propose that future researches of affective temperament should utilize neuroimaging, along with neurogenetic studies, and compare the obtained results with a control group. This measure might elucidate what kind of dopaminergic transmission, determined by *COMT*, is responsible for the pathogenesis of mood disorders in the obese population. 

In Table 4 and Table 5, we did not observe any statistically significant associations between the affective temperament and polymorphisms of *DAT1* nor *DRD4*. 

The literature shows mixed results about the connection between the abovementioned polymorphism and temperament analyzed with various scales. According to Cloninger’s theory, the dimension of temperament novelty seeking is, according to this concept, related to the *DRD4* gene. Previous studies on the association of the VNTR polymorphism in the *DRD4* gene suggested association with the dimension of novelty seeking of temperament [90]. However, further studies did not detect a similar relationship, but showed a correlation of the polymorphism (-521 C/T) of the DRD4 gene with impulsivity and novelty seeking. Other researchers have found a connection between the VNTR polymorphism and two mood temperaments: cyclothymic and irritable; however, this study was performed on a healthy volunteer of the Asian population, and therefore it may be difficult to compare the results to our group [87].

Regarding the *DAT1* gene, some studies indicate that the VNTR 3’UTR polymorphism of the *DAT1* gene is associated with novelty seeking; however, other researchers have not obtained similar results [91,92,93]. The research also indicates the interaction of *DAT1* gene polymorphisms, *DRD4* and neuroticism [94]. The literature shows little findings describing affective temperament measured with TEMPS-Am, and *DRDR4* or *DAT1* polymorphisms, and more studies are needed in this field. 

In Table 6, the effect interaction was observed for the anxious dimension and BMI. However, by using a logistic regression we have not obtained significant results for the BMI and any temperament dimension. In the study of Amann et al. (2009), obese patients scored significantly higher in the anxious dimension, as well as for the irritable and cyclothymic factors [21]. Therefore, we assume that persons characterized by an anxious temperament might be at greater risk of further weight gain. Even though we did not find any associations between this dimension and the dopaminergic genes, it could be that an anxious temperament is related to the serotonergic transmission. It could be, in particular, that it has been linked to moderate novelty seeking and greater harm avoidance—which is connected to this type of signaling [95]. Amann et al. (2009) displayed an association between the S allele of the 5HTTLPRI polymorphism in the serotonin transporter gene and greater scores in the following TEMPS dimensions: cyclothymic, irritable and anxious [21]. Gonda et al. (2006) also obtained similar results in the group of women, which indicates the relationship between an affective temperament and the serotonergic transmission [96]. Additionally, in our previous study regarding the *5HTTLPR* polymorphism, subjects homozygous to the S allele exhibited higher scores in anxious and depressive dimensions in comparison to L allele carriers. Such results indicate a stronger connection between the affective temperament measured by TEMPS-A and the serotonergic transmission, instead of dopaminergic signaling in the obese population [23]. 

In this study we analyzed only one neurotransmitter signaling. We must take into consideration that many factors influence behavior, including other gene polymorphisms or the complex neurotransmitter interactions in different brain areas [96,97,98,99,100,101]. For instance, functional brain imaging revealed an additive effect of *COMT* Met158 and *5-HTTLPR* S alleles on the response of the amygdale, hippocampal and limbic cortical areas to unpleasant stimuli, suggesting that persons with those alleles may show a lowered resilience against an anxiety mood [101]. An interesting study of Ro et al. (2018) indicates the differences in the expression of glucagon-like peptide 1 and 2 receptors (GLP-!R, GLP-2R) in patients suffering from mood disorders in comparison to healthy controls, with a greater susceptibility connected to higher BMI values. Both GLP-1R and GLP-2R are implicated in neuroprotection and the antidepressant effect [102]. Moreover, it has been found that a lower expression of the leptin receptor in the hippocampus and hypothalamus may have a significant impact on obesity and comorbid depression. Researchers found that obese individuals or those exposed to chronic unpredictable mild stress showed a diminished expression of the leptin receptor [103]. Nonetheless, mood disorders are complex in their nature and constitute a hard challenge for clinicians in their practice. Due to the growing problem of obesity, there is a need for creating more effective preventing programs that tackle the occurrence of affective disorders in this population. Hence, more studies focusing on the molecular basis of the pathogenesis and interplay between both disorders could bring a better understanding, which is essential for predicting the course and nature of the diseases.

## 4. Materials and Methods

### 4.1. Participants

The study was conducted on a population of 245 Caucasian people, who were diagnosed with primary obesity. Secondary causes of obesity were excluded in the Clinic of Endocrinology and Diabetology at the Collegium Medicum of the Nicolaus Copernicus in Bydgoszcz on the basis of a subjective and objective medical assessment, as well as on the basis of performed hormonal and metabolic tests. Significant physical diseases, addiction, substance abuse (e.g., cannabis misuse) or psychiatric and neurological illnesses were the excluding factors for participation in the study. All patients, after being given detailed information on the purpose and nature of the study, expressed written and informed consent of their participation. The study obtained the consent of the bioethical commission at the Nicolaus Copernicus University (No 533/ 2008, 15 Dec 2008).

### 4.2. Clinical Assessments and Measures 

Building on the assessed anthropometric factors, the diagnosis of obesity was established. As a factor reflecting the amount of body fat, the BMI index was adopted. It was calculated as the ratio of weight (kg) to square of height (m).

### 4.3. Psychological Assessment

For the psychological assessment, we utilized the Temperament Evaluation of Memphis, Pisa, Paris and San Diego Autoquestionnaire (TEMPS-A) to perform an analysis of the dimensions of the affective temperament.

### 4.4. Genotyping 

Genomic DNA was obtained from peripheral blood (5 mL) using the method developed by Lahiri and Schnabel (1993) [104]. The blood was collected on the EDTA medium and mixed, before being frozen in liquid nitrogen and stored at −80 °C prior to extraction. The polymorphisms of the *DAT1, COMT* and *DRD4* genes were determined using the polymerase chain reaction (PCR). The following primers were used: *DAT1* forward, 5′-TGTGGTGTAGGGAACGGCCTGAG-3′; *DAT1* reverse, 5′-CTTCCTGGAGGTCACGGCTCAAGG-3′; forward, 5′-AGCTCCAAGCGCGCTCACAG-3′; *COMT* reverse, 5′-CAAAGTGCGCATGCCCTCCC-3′; DRD4 forward: 5′-GCGACTACGTGGTCTACTCG-3′; and *DRD4* rewers: 5′-AGGACCCTCATGGCC TTGC-3′. The PCR products were then separated by agarose gel electrophoresis using O’RangeRuler™ 50 bp DNA Ladder (Fermentas) as a length marker (Figure 1, Figure 2 and Figure 3).

### 4.5. Statistical Analysis

Using the Shapiro-Wilk test, it was determined that the test group does not meet the normal distribution criteria. The statistical significance of the differences between the two groups was calculated using the Mann–Whitney U test, and for comparisons with three or more groups, the Kruskal–Wallis analysis of variance (ANOVA) was applied. The NIR Fisher test was used for post hoc analyses. Correlations between two quantitative variables were examined using the Spearman rank correlation test. To control for the effect of age and BMI, which both exhibit significant simple correlations with the dimensions of temperament, we analyzed the data with a partial Kendall regression (partial Kendall’s Tau), the nonparametric technique that controls for one confounding [105].

An analysis of covariance (ANCOVA) was performed to examine the interaction effects. An effect size was determined using Cohen’s d. The gathered data were analysed by means of StatSoft, Inc. (2017) using Statistica, version 13.0 software and the computer program “Utility Programs for Analysis of Genetic Linkage” (Copyright © 1988 J. Tot) was utilized to test for the goodness of fit to the Hardy–Weinberg equilibrium. The distributions of all three analyzed genotypes were against the Hardy–Weinberg equilibrium.

Bonferroni corrections were used as multiple testing procedures. A logistic regression of data was performed to predict logit on TEMPS-A temperaments subscales (The Wald statistic in Logistic regression model).

## 5. Conclusions

To our knowledge, this is the first study analyzing the affective temperament in an obese population in the context of dopaminergic genes polymorphisms, including *COMT* Val158Met, *DRD4*, and *DAT1*. The results of our study indicate the connection between the irritable and cyclothymic dimensions in *COMT* heterozygotes only. We presume that the dopaminergic transmission modulated by these COMT gene polymorphisms may entail a significant expression of cyclothymic and irritable temperaments. This is a very interesting finding, giving rise to more sophisticated research in the future, utilizing neuroimaging studies. 

## 6. Limitations

The main limitation of our study is the lack of a control group in order to gain more reliable results. Second, for the proper evaluation of the connection between the affective temperament and gene polymorphisms, our study group should be larger. 

## Figures and Tables

**Figure 1 ijms-20-01847-f001:**
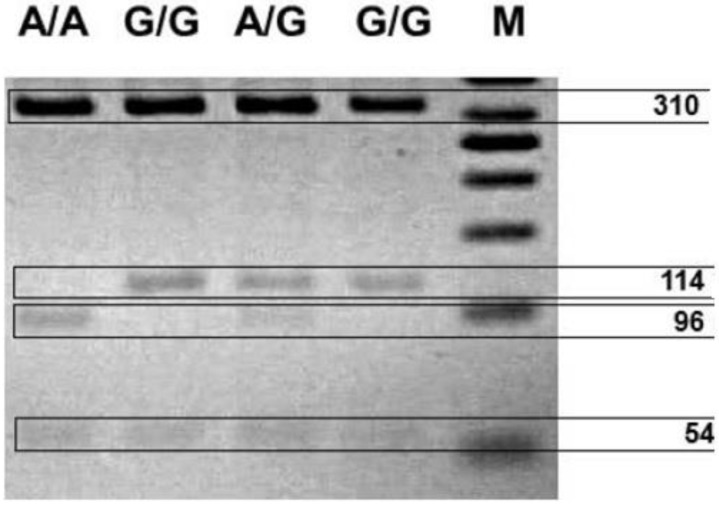
Photo of the digested *COMT* PCR products. The results are labeled by genotype: Met/Met (A/A), 96 bp only; Val/Met (A/G) 114 and 96 bp; and Val/Val (G/G), 114 bp only.

**Figure 2 ijms-20-01847-f002:**
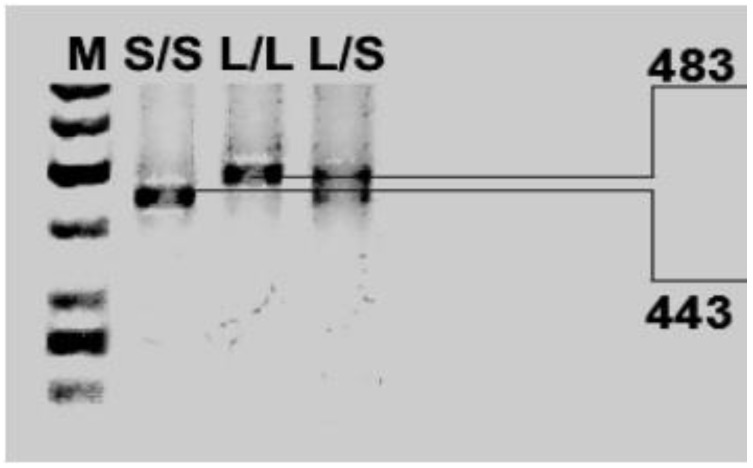
Photo of digested *DAT1* PCR products. The results are labeled by genotype: 10/10 (L/L) 483 bp only; 10/9 (L/S) 483 and 443 bp; and 9/9 (S/S) 443 bp only.

**Figure 3 ijms-20-01847-f003:**
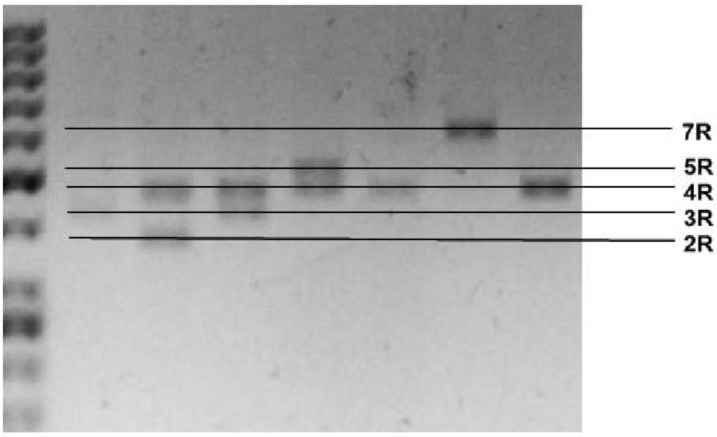
Photo of digested *DRD4* PCR products. Representative photo of separated *DRD4* PCR products depending on the genotype: LL—only 619 bp band (7R); S/S 379 bd (2R) or/and 427 bp (3R) or/and 523 bp (5R) band; L/S – 379 bd (2R) or 427 bp (3R) or 523 bp (5R) and 619 bp (7R) bands.

**Table 1 ijms-20-01847-t001:** Age, body mass index (BMI) and results on the TEMPS-A scale in study participants. Data are presented as medians, and 25th and 75th quartiles.

	Female	Male	P	Cohen’s d
	(*n* = 178)	(*n* = 67)		
Age	41	42	0.11	0.24
	(36.0–47.0)	(34.0–48.5)		
BMI	40.7	41.4	0.8	0.03
	(36.3–47.0)	(35.2–48.5)		
TEMPS_D	0.38	0.43	0.04	0.36
	(0.28–0.52)	(0.24–0.43)		
TEMPS_C	0.36	0.47	0.09	0.29
	(0.24–0.52)	(0.23–0.62)		
TEMPS_H	0.52	0.52	0.53	0.1
	(0.35–0.62)	(0.38–0.64)		
TEMPS_I	0.14	0.23	0.001	0.42
	(0.05–0.28)	(0.09–0.33)		
TEMPS_A	0.33	0.35	0.12	0.18
	(0.24–0.55)	(0.17–0.51)		

BMI, body mass index; TEMPS_D—depressive subscale of TEMPS-A; TEMPS_C—cyclothymic subscale of TEMPS-A; hyperthymic subscale of TEMPS-A; TEMPS_I—irritable subscale of TEMPS-A; TEMPS_A—anxious subscale of TEMPS-A. Significance of differences between sexes was determined by the Mann—Whitney U test. Size effect was measured by Cohen’s d method.

**Table 2 ijms-20-01847-t002:** R-Spearman correlations of the age and BMI result with the TEMPS scores in women and men. Partial Kendall regression for significant correlations.

	Female		Male	
	(*n* = 178)		(*n* = 67)	
	Age	BMI	Age	BMI
TEMPS_D	r = 0.21			
	*p* = 0.004	r = −0.14	r = −0.05	r = 0.06
	Par. Kendall’s tau	*p* = 0.06	*p* = 0.68	*p* = 0.63
	Tau = -2.74; *p* = 0.006			
TEMPS_C		r = −0.16	r = −0.26	r = 0.33
	r = −0.06	*p* = 0.03	*p* = 0.03	*p* = 0.006
	*p* = 0.42	Par. Kendall’s tau	Par. Kendall’s tau	Par. Kendall’s tau
		Tau = −0.15; *p* = 0.002	Tau = −0.01; *p* = 0.44	Tau = −0,24; *p* = 0.003
TEMPS_H		r = 0.16		
	r = −0.09	*p* = 0.03	r = 0.09	r = −0.09
	*p* = 0.23	Par. Kendall’s tau	*p* = 0.46	*p* = 0.47
		Tau = 0.01; *p* = 0.37		
TEMPS_I				r = 0.31
	r = 0.004	r = −0.07	r = −0.12	*p* = 0.01
	*p* = 0.95	*p* = 0.35	*p* = 0.33	Par. Kendall’s tau
				Tau = −0.13; *p* = 0.05
TEMPS_A	r = 0.16			
	*p* = 0.03	r = −0.11	r = −0.11	r = 0.07
	Par. Kendall’s tau	*p* = 0.14	*p* = 0.37	*p* = 0.57
	Tau = −0.09; *p* = 0.03			

BMI, body mass index; TEMPS_D—depressive subscale of TEMPS-A; TEMPS_C—cyclothymic subscale of TEMPS-A; TEMPS_H—hyperthymic subscale of TEMPS-A; TEMPS_I—irritable subscale of TEMPS-A; TEMPS_A—anxious subscale of TEMPS-A. Par. Kendall’s tau—partial Kendall’s tau. Bold values indicate statistical significance.

**Table 3 ijms-20-01847-t003:** COMT polymorphisms and TEMPS results in study group.

	All Group				
	(*n* = 245)				
	G/G	G/A	A/A	*p*	
	(*n* = 64)	(*n* = 120)	(*n* = 61)		
BMI	40.9	42,5	42,4	0.52	
	(36.7–44.3)	(36.5–49.0)	(37.0–48.1)		
TEMPS_D	0.36	0.42	0.38	0.36	
	(0.28–0.42)	(0.28–0.52)	(0.28–0.43)		
TEMPS_C				0.04	Post-hoc
	0.28	0.47	0.38		G/G vs G/A *p* = 0.014
	(0.16–0.47)	(0.24–0.64)	(0.23–0.52)		G/A vs A/A ns
					G/Avs AA na
TEMPS_H	0.57	0.47	0.57	0.07	
	(0.50–0.67)	(0.28–0.61)	(0.38–0.57)		
TEMPS_I				0.01	Post-hoc
	0.09	0.26	0.09		G/G vs G/A *p* = 0.01
	(0.04–0.16)	(0.09–0.33)	(0.05–0.24)		G/A vs A/A ns
					G/Avs AA ns
TEMPS_A	0,32	0,35	0.32	0.52	
	(0.20–0.52)	(0.22–0.59)	(0.24–0.52)		

BMI, body mass index; TEMPS_D—depressive subscale of TEMPS-A; TEMPS_C—cyclothymic subscale of TEMPS-A; TEMPS_H—hyperthymic subscale of TEMPS-A; TEMPS_I—irritable subscale of TEMPS-A; TEMPS_A—anxious subscale of TEMPS-A. Significance of differences between subgroups was determined by the Kruskal-Wallis ANOVA. Post-hoc analysis was conducted with Fisher’s NIR test.

**Table 4 ijms-20-01847-t004:** DAT polymorphisms and TEMPS-A scale results in study group.

	All Group			
	(*n* =245)			
	L/L	L/S	S/S	*p*
	(*n* = 117)	(*n* = 103)	(*n* = 25)	
BMI	41.2	41.6	40.7	0.9
	(36.2–48.9)	(35.8–48.5)	(39.9–46.8)	
TEMPS_D	0.42	0.38	0.38	0.71
	(0.28–0.52)	(0.28–0.47)	(0.28–0.47)	
TEMPS_C	0.38	0.38	0.33	0.86
	(0.24–0.62)	(0.23–0.57)	(0.29–0.48)	
TEMPS_H	0.52	0.52	0.57	0.87
	(0.36–0.61)	(0.38–0.62)	(0.38–0.62)	
TEMPS_I	0.19	0.14	0.09	0.23
	(0.07–0.33)	(0.05–0.28)	(0.04–0.29)	
TEMPS_A	0.32	0.33	0.44	0.41
	(0.21–0.52)	(0.24–0.52)	(0.28–0.59)	

BMI, body mass index; TEMPS_D—depressive subscale of TEMPS-A; TEMPS_C—cyclothymic subscale of TEMPS-A; TEMPS_H—hyperthymic subscale of TEMPS-A; TEMPS_I—irritable subscale of TEMPS-A; TEMPS_A—anxious subscale of TEMPS-A. Significance of differences between subgroups was determined by the Kruskal-Wallis ANOVA.

**Table 5 ijms-20-01847-t005:** DRD4 polymorphisms and TEMPS-A results in subgroups of women and men.

	All Group		
	(*n* = 245)		
	L/L; L/S	S/S 114	*p*
	(*n* = 84)	(*n* = 161)	
BMI	42.9	41.8	0.21
	(38.5–49.0)	(37.2–47.1)	
TEMPS_D	0.4	0.33	0.25
	(0.28–0.47)	(0.28–0.47)	
TEMPS_C	0.38	0.47	0.64
	(0.24–0.61)	(0.23–0.57)	
TEMPS_H	0.47	0.19	0.15
	(0.35–0.59)	(0.05–0.28)	
TEMPS_I	0.16	0.19	0.27
	(0.05–0.33)	(0.20–0.55)	
TEMPS_A	0.32	0.35	0.75
	(0.24–0.47)	(0.20–0.54)	

BMI, body mass index; TEMPS_D—depressive subscale of TEMPS-A; TEMPS_C—cyclothymic subscale of TEMPS-A; TEMPS_H—hyperthymic subscale of TEMPS-A; TEMPS_I—irritable subscale of TEMPS-A; TEMPS_A—anxious subscale of TEMPS-A. Significance of differences between subgroups was determined by the Kruskal-Wallis ANOVA.

**Table 6 ijms-20-01847-t006:** Analyses of unidimensional interaction effects for TEMPS-A temperaments subscales.

	TEMPS-D			TEMPS-C			TEMPS-H			TEMPS-I			TEMPS-A		
	SS	F	*p*	SS	F	*p*	SS	F	*p*	SS	F	*p*	SS	F	*p*
Gender	0.15	5.3	0.02	0.23	4.6	0.03	0.01	0.36	0.54	0.18	6.1	0.01	0.04	0.94	0.33
Age	2.08	1.38	0.06	2.07	0.68	0.94	2.07	1.01	0.46	1.89	1.19	0.20	3.03	1.17	0.22
BMI	0.11	0.78	0.65	0.35	4.7	0.11	0.06	0.20	0.96	0.37	4.9	0.10	0.24	17.1	0.01
DAT1	0.02	0.38	0.68	0.22	0.22	0.79	0.007	0.1	0.90	0.09	1.53	0.21	0.05	0.52	0.59
COMT	0.07	1.49	0.22	0.32	3.2	0.04	0.22	2.9	0.05	0.20	3.4	0.03	0.10	1.07	0.34
DRD4	0.02	0.49	0.61	0.04	0.39	0.67	0.07	1.08	0.34	0.07	1.36	0.35	0.01	0.11	0.89

One-dimensional analysis of significance (ANOVA) F-test based on SS.

**Table 7 ijms-20-01847-t007:** Logistic regression model coefficients on TEMPS-A temperaments subscales.

	**TEMPS-D**						
	**B**	**S.E.**	**Wald**	**df**	***p***	**95%C.I.**	**95%C.I.**
						**Lower**	**Upper**
Gender	0.164	0.057	8.05	1	0.004	0.278	0.05
Age	0.003	0.004	0.69	1	0.4	0.011	−0.004
BMI	0.00008	0.006	0.01	1	0.89	0.013	−0.11
DAT1	0.036	0.06	0.37	2	0.82	0.157	−0.085
COMT	−0.126	0.07	5.7	2	0.057	0.018	−0.272
DRD4	−0.100	0.196	0.27	2	0.86	0.284	−0.485
	**TEMPS-C**						
	**B**	**S.E.**	**Wald**	**df**	***p***	**95%C.I.**	**95%C.I.**
						**Lower**	**Upper**
Gender	−0.13	0.066	3.95	1	0.04	−0.001	−0.262
Age	−0.0007	0.005	0.01	1	0.9	0.01	−0.012
BMI	0.008	0.008	1.09	1	0.29	0.296	−0.007
DAT1	−0.045	0.08	0.3	2	0.85	0.12	−0.21
COMT	−0.177	0.111	2.99	2	0.055	0.04	−0.39
DRD4	0.137	0.168	0.83	2	0.65	0.366	0.117
	**TEMPS-H**						
	**B**	**S.E.**	**Wald**	**df**	***p***	**95%C.I.**	**95%C.I.**
						**Lower**	**Upper**
Gender	−0.07	0.05	2.33	1	0.12	0.021	−0.175
Age	0.0008	0.003	0.05	1	0.81	0.008	−0.006
BMI	0.002	0.006	0.2	1	0.64	0.014	−0.009
DAT1	−0.019	0.06	0.27	2	0.87	0.098	−0.137
COMT	0.12	0.06	6.05	2	0.04	0.241	−0.002
DRD4	−0.07	0.18	2.45	2	0.29	0.282	−0.239
	**TEMPS-I**						
	**B**	**S.E.**	**Wald**	**df**	***p***	**95%C.I.**	**95%C.I.**
						**Lower**	**Upper**
Gender	0.032	0.094	0.11	1	0.73	0.217	−0.152
Age	0.003	0.008	0.17	1	0.67	0.021	−0.013
BMI	0.005	0.014	0.13	1	0.71	0.033	−0.023
DAT1	0.299	0.143	5.44	2	0.065	0.58	0.019
COMT	−0.35	0.211	5.92	2	0.04	0.074	−0.756
DRD4	0.322	0.207	3.13	2	0.2	−0.084	0.12
	**TEMPS-A**						
	**B**	**S.E.**	**Wald**	**df**	***p***	**95%C.I.**	**95%C.I.**
						**Lower**	**Upper**
Gender	0.085	0.078	1.18	1	0.27	0.238	−0.068
Age	0.005	0.005	0.8	1	0.37	0.016	−0.005
BMI	−0.008	0.008	0.95	1	0.32	0.008	−0.026
DAT1	−0.08	0.086	1.05	2	0.59	0.349	0.088
COMT	−0.219	0.1	4.2	2	0.12	0.04	−0.009
DRD4	−0.11	0.257	0.21	2	0.89	0.386	−0.623

BMI, body mass index; TEMPS_D—depressive subscale of TEMPS-A; TEMPS_C—cyclothymic subscale of TEMPS-A; TEMPS_H—hyperthymic subscale of TEMPS-A; TEMPS_I—irritable subscale of TEMPS-A; TEMPS_A—anxious subscale of TEMPS-A.

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
