# Peer review of "The Association Between Affective Temperament Traits and Dopamine Genes in Obese Population"

_ijms, 2019, doi:10.3390/ijms20081847_

Round 1

Reviewer 1 Report

I have reviewed this manuscript multiple times. I am glad to see that the authors have improved on their statistical analyses. I have no further comments.

Reviewer 2 Report

I recommend publication.

This manuscript is a resubmission of an earlier submission. The following is a list of the peer review reports and author responses from that submission.

Round 1

Reviewer 1 Report

While the reviewers have attempted to improve the manuscript, the study description and results are still confusing with the presented results not confirming their conclusions.

 Description of the statistical analyses are lacking and the presented analyses are frequently inadequately applied. On line 118-119 the authors stated that they performed a correlation analysis between COMT genotypes and diagnostic subgroups of both male and female subjects. Then despite of a lack of main effect, they performed post-hoc test!? This is wrong. The authors did not performed a post-hoc test, they performed a multiple comparison test.  Post-hoc test should be performed only after a significant main effect has been observed. However, that is not what they observed or reported.  

From the initial analyses presented in table 2 it seems that female subjects indeed have a greater correlation with the depression subscale, however, the main research hypothesis is to investigate the relationship between BMI and temperament, not temperament and age. Ideally, they should have tested the relationship between BMI and temperament adjusting for age.   

Author Response

Description of the statistical analyses are lacking and the presented analyses are frequently inadequately applied. On line 118-119 the authors stated that they performed a correlation analysis between COMT genotypes and diagnostic subgroups of both male and female subjects. Then despite of a lack of main effect, they performed post-hoc test!? This is wrong. The authors did not performed a post-hoc test, they performed a multiple comparison test.  Post-hoc test should be performed only after a significant main effect has been observed. However, that is not what they observed or reported.

Thank you for this important consideration. In fact, in the description of statistical analyzes relating to Table 3, there was inaccuracy. What we did in turn was, first of all, an attempt to determine differences in the intensity of temperaments in the context of genotypes in subgroups of women and men. Due to the fact that the results were not significant, we moved to the second stage. At this stage, we examined the intensity of temperaments in the context of genotypes in the whole group. And only for the differences significant for Kruskal-Willis ANOVA, in order to determine the significance of differences between specific groups, we performed a post hoc analysis using the NIR test. And this is what describes the contents of table 3. We have, of course, changed the description referring to this remark.

From the initial analyses presented in table 2 it seems that female subjects indeed have a greater correlation with the depression subscale, however, the main research hypothesis is to investigate the relationship between BMI and temperament, not temperament and age. Ideally, they should have tested the relationship between BMI and temperament adjusting for age.  

This is also an important point. Initially, we did not want to develop the analyzes included in table 2 because it was not the main goal of the work. Nevertheless, after this suggestion, we conducted a further analysis, using partial Kendall regression, of the data and we included its results in Table 2.

Reviewer 2 Report

In the revised manuscript, the authors addressed most of the major questions raised by Reviewers improving both the main structure and quality of the present paper. I have no further additional comments.

Author Response

Thank you for your comments. 

Reviewer 3 Report

Thank you for inviting me to review the paper on “The association between affective temperament traits 3 and dopamine genes in obese population.” As there are track changes in the manuscript, I assume this paper went through one round of revision. I think this is an important paper and the study was well conducted. I have the following recommendations:

1.    In line 37, the authors stated that “Recent psychiatric studies suggest that there is a link between obesity, mood disorders, in particular depression and bipolar disorder [3,4].” It is important to state that such relationship started in childhood. Please add the following statement and modify the subsequent sentence:

…. in particular depression and bipolar disorder [3,4].”  The association between obesity and depression occurred in childhood (Reference: PMID: 28401646). Previous research indicated that the symptoms of eating disorders are common and that patients with bipolar disorder are more obese than the control group.

2.    I cannot find any exclusion criteria on substance abuse (e.g. cannabis use) as cannabis use may be associated with COMT genotype, DRD4 genotype and obesity. It would be helpful if the authors can clarify the prevalence of cannabis misuse among the participants. If such information is not available, please state in the limitation.

3.     In line 373, the authors stated that “including other gene polymorphisms or the complex neurotransmitters interactions in different brain areas”. I suggest the authors to name some specific examples. Please mention the findings of the effect of body mass index on glucagon-like peptide receptor gene expression in the post mortem brain from individuals with mood disorder and the relationship between chronic unpredictable mild stress, depression and leptin by referring to most recent studies in these areas.

Author Response

In line 37, the authors stated that “Recent psychiatric studies suggest that there is a link between obesity, mood disorders, in particular depression and bipolar disorder [3,4].” It is important to state that such relationship started in childhood. Please add the following statement and modify the subsequent sentence:

…. in particular depression and bipolar disorder [3,4].”  The association between obesity and depression occurred in childhood (Reference: PMID: 28401646). Previous research indicated that the symptoms of eating disorders are common and that patients with bipolar disorder are more obese than the control group.

Due to reviewer’s suggestion, we added the proposed sentence and modified the sentence from the line 37. We also cited recommended referenece.

I cannot find any exclusion criteria on substance abuse (e.g. cannabis use) as cannabis use may be associated with COMT genotype, DRD4 genotype and obesity. It would be helpful if the authors can clarify the prevalence of cannabis misuse among the participants. If such information is not available, please state in the limitation.

Individuals using cannabis were excluded from the participation of this study. Hence, we modified exclusion criteria within Methodology – participants addicted to, or misuing any psychoactive substance were excluded from the study.

In line 373, the authors stated that “including other gene polymorphisms or the complex neurotransmitters interactions in different brain areas”. I suggest the authors to name some specific examples.

The specific gene x gene example between COMTVal158Met and 5HTTLPR interaction has been mentioned within Discussion section.

Please mention the findings of the effect of body mass index on glucagon-like peptide receptor gene expression in the post mortem brain from individuals with mood disorder and the relationship between chronic unpredictable mild stress, depression and leptin by referring to most recent studies in these areas.

We cited and briefly discussed suggested studies within Discussion section.

Round 2

Reviewer 1 Report

I'm afraid that after reading the revised manuscript I'm still not convinced in the validity of the results presented by the authors.  Briefly, as I have stated in my previous comments, the ideal approach for the authors to test their data is to build a generalized linear regression model for the entire sample, adjust for any potential confounding and present the results from the test. I'm afraid that their piecemeal analysis that resulted from artificial splitting of the sample is not warranted and really seems to be a fishing expedition, the goal of which, is to find significant results. Furthermore, as I have also suggested in my previous comments, ideally the authors should have first determined whether any of the tested polymorphisms are associated with BMI and if so then ask about the mediating effect of obesity on the psychological profile of the studied subjects. Currently, the research design presented by the authors is merely asking whether there are significant correlations between dopaminergic variants and various temperament behaviors in obese people. As such research design is flawed presenting itself with a complete confounding, as they do not report a contrasting sample of non-obese subjects.     To that end I still think that revised manuscript is of insufficient quality to be published, at least not until the research design, statistical analyses and research questions are completely redesigned.